# Legal education reform and medical litigation: Improved access but delayed justice in plastic surgery malpractice cases in South Korea

Daihun Kang[1,2], Seung Eun Hong[1,2]*

1 Department of Plastic and Reconstructive Surgery, Ewha Womans University Seoul Hospital, Seoul, Republic of Korea, 2 College of Medicine, Ewha Womans University, Gangseo-gu, Seoul, Republic of Korea

* monkeyhong@hanmail.net

## Abstract

South Korea's 2009 legal education reform, which replaced the bar exam and two-year judicial training system with graduate-level law schools, aimed to enhance the quality and accessibility of legal services. This study examines its impact on malpractice litigation in plastic and reconstructive surgeries and procedures (PRSP). Using publicly available civil court decisions, we conducted a retrospective analysis comparing litigation patterns before (2006–2012, n = 23) and after (2017–2021, n = 75) the reform, excluding the transitional period (2013–2017). Our findings reveal significant increases in litigation frequency (3.28 to 15 cases annually, p < 0.001) and case duration (median: 969–1,570 days, p < 0.001). While the increased litigation frequency and reduced adjudicated damages (median reduction: 75.8%) suggest improved accessibility to legal services - aligning with reform objectives - prolonged case durations highlight unresolved systemic challenges. The transition from practical training to academic-focused legal education, coupled with limited expansion of judicial infrastructure, has contributed to procedural delays despite consistent judicial standards in malpractice assessments. These findings underscore the dual impacts of the reform, achieving partial success in improving access to legal representation but exposing inefficiencies in judicial capacity. To address these challenges, short-term measures such as appointing additional judges and establishing specialized court divisions are needed. In the mid-term, implementing AI-based case management systems could enhance procedural efficiency. Long-term strategies, including structured mediation programs, are critical to alleviating court burdens and ensuring timely resolutions.

**Data availability statement:** All relevant data are within the paper and its Supporting Information files.

**Funding:** The author(s) received no specific funding for this work.

**Competing interests:** The authors have declared that no competing interests exist.

**Abbreviations:** AI, Artificial Intelligence; PRSP, Plastic and Reconstructive Surgeries and Procedures; COVID-19, Coronavirus Disease 2019; KRW, Korean Won; USD, United States Dollar; SPSS, Statistical Package for the Social Sciences; n = Number; IQR, Interquartile Range; SD, Standard Deviation; CI, Confidence Interval; U, U statistic (The U statistic is the test statistic for the Mann-Whitney U test.); Z, Z - score (The Z-score represents the standardized test statistic, which is a transformation of the U statistic.); p, p – value (Probability Value).

## Introduction

"Justice delayed is justice denied."

This principle underscores the critical role of judicial efficiency in safeguarding access to justice [1]. In 2009, South Korea introduced a graduate-level law school system, replacing its traditional bar exam and two-year judicial training model, to enhance legal expertise and expand access to legal services [2]. The previous system, which required successful bar exam candidates to complete a two-year practical training program at the Judicial Research and Training Institute [3], while producing competent legal professionals, faced criticism for its limited capacity, exclusivity, and inability to address the needs of an increasingly complex and diversified society [4]. The reform aimed to address these challenges by increasing the number of legal professionals, diversifying their educational and professional backgrounds, and equipping them with the expertise necessary to navigate the complexities of modern legal disputes [5].

However, well-intentioned reforms can sometimes yield unintended consequences. Just as Australia's introduction of foxes to control rabbit populations inadvertently disrupted native ecosystems [6], South Korea's legal education reform may have introduced challenges such as shifts in litigation patterns and strains on judicial efficiency. The principle "more lawyers, more lawsuits, longer delays" reflects how an increased supply of legal professionals can lower barriers to litigation while simultaneously overburdening an underprepared judicial system [7,8]. Despite these challenges, empirical studies analyzing the real-world impacts of South Korea's legal education reform remain scarce, particularly in relation to judicial efficiency and litigation behaviors.

This study addresses this gap by focusing on civil court decisions in plastic and reconstructive surgery and procedures (PRSP), a specialized domain where South Korea holds global prominence. PRSP litigation is characterized by its high volume and the complexity of associated legal disputes, making it an ideal lens through which to examine the systemic effects of legal education reform. By comparing litigation patterns before and after the reform, this research investigates whether the introduction of law schools has led to "more lawsuits" and "longer delays."

Beyond its national scope, this study contributes to the global discourse on legal education reforms by drawing comparative insights from international contexts. For instance, Japan's parallel reforms faced similar challenges in balancing legal expertise with judicial efficiency [9], while China's gradual improvements highlight the benefits of sustained investments in judicial capacity [10]. Germany's well-established mediation systems, such as Gutachterkommissionen and Schlichtungsstellen [11], serve as a model for integrating alternative dispute resolution mechanisms to alleviate judicial burdens. By situating South Korea's experience within this broader framework, this research underscores the importance of aligning education policies with judicial capacity and societal needs. Specifically, the transition from a bar exam-based system—where successful candidates underwent a two-year practical training program at the Judicial Research and Training Institute before

being appointed as judges, prosecutors, or private attorneys based on their performance - to a new graduate-level law school system in South Korea highlights the critical role of education policies in shaping the competencies of legal professionals. While the new system emphasizes theoretical knowledge and bar exam preparation, the absence of mandatory practical training raises concerns about the readiness of graduates to handle complex litigation, such as medical malpractice cases. Drawing from international experiences, this study emphasizes the need for education policies that strike a balance between academic rigor and practical skills, offering actionable lessons for policymakers and legal scholars worldwide.

## Methods

### Study design and data collection

This study did not involve clinical trials. Clinical trial number: not applicable.

This retrospective study analyzed civil court decisions related to PRSP malpractice in South Korea, focusing on changes before and after the introduction of law school graduates into the legal profession, including both attorneys and judges. Data were collected from multiple legal databases, including the Supreme Court of Korea's case database [12], LAWnB [13], BigCase [14], and Casenote [15]. The search strategy utilized specific terms such as "cosmetic surgery," "plastic surgery," "reconstructive surgery," "filler," "cosmetic procedure," "Botox," "liposuction," "breast augmentation," "cosmetic malpractice," "cosmetic surgery complications," "plastic surgery complications," and "medical disputes."

### Study period and sample

The study period was strategically divided into two segments to capture the impact of legal education reform:

- **Pre-Law School Graduate Period:** January 1, 2006 – December 31, 2012

- **Post-Law School Graduate Period:** March 1, 2017 – December 31, 2021

The post-law school graduate period was limited to December 31, 2021, to avoid potential biases introduced by the Coronavirus disease - 2019 (COVID-19) pandemic. During the pandemic, significant disruptions in court operations led to delays in legal proceedings, which could have skewed the data. Excluding this period helps maintain the integrity of the study by focusing on data unaffected by these external disruptions.

### Inclusion criteria

Cases were included if they met the following criteria:

1. Civil lawsuits related to complications or dissatisfaction following plastic and reconstructive surgeries and procedures (PRSP)

2. First-instance court decisions

3. Plaintiffs seeking damages for alleged medical malpractice

4. Cases involving allegations of breach of duty of care or duty to inform

### Exclusion criteria

To ensure relevance and consistency, the following cases were excluded:

1. Cases dated between January 1, 2013, and February 28, 2017, due to:

- The requirement for law school graduates to gain 6 months of practical experience before establishing or joining a law practice (as stipulated in Article 21-2 of the Attorney-at-Law Act)

- Our study deliberately excludes the period of 2013 - 2017 to ensure a rigorous and unbiased assessment of the reform's mature impact. This period represents a transitional phase during which law school graduates were still integrating into the judiciary and legal profession, undergoing unique institutional adjustment processes. These include the mandatory 6-month practical experience required under Article 21-2 of the Attorney-at-Law Act and the 8-month judicial training program at the Judicial Research and Training Institute, which began in 2015. During this time, new judges and lawyers were likely influenced by senior colleagues' established practices, leading to a potential blending of pre-reform and post-reform characteristics [16].

Including cases from this transitional period risks introducing confounding factors that could obscure the true effects of the reform. For instance, early law school graduates may not have fully developed the professional autonomy or decision-making frameworks necessary to reflect the reform's substantive impact. Additionally, transitional dynamics, such as variations in legal strategies and adaptation challenges, could bias the results by amplifying noise within the dataset.

Furthermore, the number of law school graduates actively participating in the judiciary and legal profession during this period remained relatively low, suggesting their influence on the overall legal system was likely minimal. Including cases from this transitional period risks introducing confounding factors that could obscure the true effects of the reform.

By focusing on cases after 2017, we isolate the effects of the reform when the system had reached a state of equilibrium. At this stage, law school graduate judges and lawyers had established their professional independence, and the judiciary had adjusted to the influx of new legal professionals. This approach minimizes selection bias and ensures that the analysis captures the reform's long-term and stable effects, rather than temporary adjustment phenomena.

2. Cases involving three or more types of procedures, to maintain focus on specific PRSP

3. Cases involving non-medical professionals (e.g., dentists, traditional Korean medicine practitioners, unlicensed individuals)

4. Cases where the defendant was a manufacturer rather than a medical professional

5. Small claims court decisions without documented reasoning

- In the South Korean legal system, small claims court decisions (currently set at 30 million KRW, approximately 23,000 USD) do not require written explanations of the judgment rationale, as stipulated in the Small Claims Act [17]. This procedural simplification aims to expedite proceedings for claims under a certain monetary threshold. While these cases constitute a portion of medical disputes, they were necessarily excluded from our analysis as they lack the detailed legal reasoning essential for examining judicial decision-making patterns and the qualitative aspects of legal interpretation that are central to our research questions.

6. Non-civil cases and settled cases

## Data analysis

Statistical analyses were performed using the Statistical Package for the Social Sciences (SPSS) version 29.0.2.0 (IBM Corp., Armonk, NY, USA) to assess differences between the pre- and post-law school graduate periods in PRSP malpractice litigation. The following analyses were conducted:

- **Shapiro-Wilk Test**: This test was used to assess the normality of continuous variables to determine the appropriate statistical methods. Prior to selecting statistical tests, we conducted comprehensive normality testing of all continuous variables. The distribution of cases across geographical regions (capital vs. provincial areas) was found to satisfy normality, allowing for the use of parametric tests such as the Chi-square test. However, other continuous variables including adjudicated

damages, consolation money, percentage of liability assigned to physicians, and the duration from the incident to the court decision showed non-normal distributions, justifying the use of non-parametric tests for these variables.

- **Mann-Whitney U Test**: Employed to compare non-normally distributed continuous variables between the two periods, such as adjudicated damages, consolation money, percentage of liability assigned to physicians, and case duration. This non-parametric test was specifically chosen over parametric alternatives because it does not assume normal distribution and is robust for comparing two independent groups with different sample sizes, making it particularly suitable for our dataset structure.

- **Chi-Square Test**: Conducted to examine the association between the introduction of the law school system and the distribution of cases across different geographical regions (capital vs. provincial areas), testing for shifts in the location of litigation. This test was appropriate as the geographical distribution data met the assumptions of normal distribution and expected cell frequencies.

- **Fisher's Exact Test**: Used to analyze categorical variables, specifically the overall plaintiff success rates between the two periods. This test was selected over the Chi-square test because it provides exact p-values and maintains statistical validity even with small sample sizes and unequal group distributions, which was particularly important given our pre-law school period sample size.

- **Fisher's Exact Test with Monte Carlo Simulation**: Applied to evaluate the association between the study periods and the types of PRSP malpractice cases. This approach was chosen because our contingency tables for procedure types contained multiple categories with some cells having expected frequencies less than 5. This simulation is a statistical method that uses repeated random sampling to obtain numerical results and estimate probabilities. It was particularly useful for analyzing sparse data where traditional statistical tests might be less reliable. By running 10,000 iterations, this method helped ensure accurate p-value estimation despite our relatively small sample sizes in some categories.

A p-value $< 0.05$ was considered statistically significant for all analyses. All statistical analyses were conducted following the reporting guidelines for observational studies and standard practices in medical litigation research.

## Ethical considerations

This study was conducted in compliance with ethical standards for research. The study protocol was reviewed and exempted by the Institutional Review Board (IRB File No. SEUMC NON2024–006). All data used were publicly available court decisions, and no personal or sensitive information was collected or analyzed.

## Use of AI tools

In this study, AI tools including Claude 3.5 Sonnet and DeepSeek-V3 were utilized for language translation. Specifically, these tools were employed to translate portions of the manuscript from Korean to English. While the AI tools provided initial translations, all final decisions regarding the content, accuracy, and refinement of the translations were made by the authors. The use of AI was strictly limited to supporting the translation process, and the authors maintained full responsibility for the final output.

## Results

1. Annual Frequency of First-Instance Civil Court Decisions Related to Plastic and Reconstructive Procedures: A Comparison Between 2006–2012 and 2017–2021

In the 2006–2012 period, 23 cases were identified, averaging 3.28 cases per year. During 2017–2021, there were 75 cases, averaging 15 cases per year (Table 1). This analysis suggests an increase in the average annual number of PRSP-related civil lawsuits in the latter period.

Statistical analysis confirmed a significant difference in the number of cases per year between the two periods (Mann-Whitney U test, U = 0.0, p < 0.001).

2. Types of Plastic and Reconstructive Procedures

The PRSPs involved in malpractice cases were classified into 12 distinct categories. This classification was based on the nature of the procedures described in the court decisions.

• Blepharoplasty (Eyelid Surgery)

• Rhinoplasty (Nose Surgery)

• Neck or Facial Lift

• Zygomatic or Mandibular Reduction Surgery

• Orthognathic Surgery

• Liposuction

• Mammoplasty or Breast Reconstruction (Breast Surgery)

• Scar Revision Surgery

• Filler and Thread Lifting

• Laser and Ultrasound Procedures

• Facial Fracture Reconstruction

• Others

**Table 1. Comparison of Case Characteristics Before and After the Introduction of Law Schools.**

| Variable | Pre-Law School Period (2006-2012) | Post-Law School Period (2017-2021) | p-value |
|---|---|---|---|
| **Number of Cases** | 23 | 75 | < 0.001 |
| **Number of Cases Per Year (mean)** | 3.28 | 15 | |
| **Plaintiff success rate (%)** | 86.95 | 81.33 | 0.755 |
| **Adjudicated Damages (KRW)** | 57,314,193 (Mean)/ 42,858 USD | 34,980,344 (Mean)/ 26,157 USD | 0.155 |
| | 12,682,749 (Median)/ 9,483 USD | 3,069,767 (Median)/ 2,295 USD | |
| **Consolation Money (KRW)** | 18,173,913 (Mean)/ 13,590 USD | 13,060,000 (Mean)/ 9,765 USD | 0.027 |
| | 15,000,000 (Median)/ 11,216 USD | 5,000,000 (Median)/ 3,738 USD | |
| **Liability Attribution (%)** | 43.6% (Mean) | 37.0% (Mean) | 0.539 |
| | 50.0% (Median) | 30.0% (Median) | |
| **Case Duration (Days)** | 1,016.74 (Mean) | 1,594.68 (Mean) | < 0.001 |
| | 969 (Median) | 1,570 (Median) | |

• KRW refers to Korean Won, the currency of South Korea. USD refers to United States Dollar, the currency of the United States. The exchange rate used for conversion was 1 USD = 1,337.30 KRW, based on the exchange rate as of August 23, 2024.

The Fisher's Exact Test with Monte Carlo Simulation revealed no statistically significant association between the period (2006–2012 vs. 2017–2021) and the type of procedures involved in malpractice cases (p = 0.216, 95% CI [0.208, 0.224]).

Despite the lack of statistical significance, we observed some notable trends in the distribution of procedures between the two periods (Table 2).

1. Blepharoplasty cases increased from 8.69% in 2006–2012 to 16% in 2017–2021.

2. Filler and thread lifting cases increased from 8.69% to 20%.

3. Rhinoplasty cases decreased from 17.39% to 8%.

4. Breast surgery cases decreased from 17.39% to 13.33%.

5. Cases involving laser and ultrasound procedures (10.66%) and facial bone fracture reconstruction (4%) emerged in the 2017–2021 period, which were absent in 2006–2012.

3. Plaintiff Success Rate

Analysis of Fisher's exact test revealed that the plaintiff success rate in the pre-law school period (2006–2012) was 86.95% (20 out of 23 cases), while in the post-law school period (2017–2021) it was 81.33% (61 out of 75 cases). Despite this slight decrease, the difference was not statistically significant (p = 0.755).

4. Adjudicated Damages

Adjudicated damages were analyzed for both periods (Table 1).

- **2006–2012 (n = 23):** The median award was 12,682,749 KRW (9,484 USD), with a mean of 57,314,193 KRW (42,858 USD). Awards ranged from 0 KRW to 340,693,767 KRW (254,762 USD), with an interquartile range (IQR) of 70,139,953 KRW (52,449 USD). The exchange rate used for conversion was 1 USD = 1,337.30 KRW, based on the exchange rate as of August 23, 2024.

- **2017–2021 (n = 75):** The median award was 3,069,767 KRW (2,295 USD), with a mean of 34,980,344 KRW (26,157 USD). Awards ranged from 0 KRW to 631,449,765 KRW (472,183 USD), with an IQR of 28,102,400 KRW (21,014 USD).

**Table 2. Types of Procedures Involved in Malpractice Cases Before and After the Introduction of Law Schools (2006-2012 vs. 2017-2021).**

| Procedure Type | Pre-Law School Period (%) | Post-Law School Period (%) | Change (%) |
|---|---|---|---|
| Blepharoplasty (Eyelid Surgery) | 8.69 | 16 | 7.31 |
| Rhinoplasty (Nose Surgery) | 17.39 | 8 | -9.39 |
| Neck or Facial Lift | 10 | 15 | 5 |
| Zygomatic or Mandibular Reduction Surgery | 5 | 7 | 2 |
| Orthognathic Surgery | 4 | 6 | 2 |
| Liposuction | 12 | 14 | 2 |
| Mammoplasty or Breast Reconstruction | 17.39 | 13.33 | -4.06 |
| Scar Revision Surgery | 3 | 5 | 2 |
| Filler and Thread Lifting | 8.69 | 20 | 11.31 |
| Laser and Ultrasound Procedures | 0 | 10.66 | 10.66 |
| Facial Fracture Reconstruction | 0 | 4 | 4 |
| Others | 13.84 | 12.01 | -1.83 |

•Fisher's exact test with Monte Carlo simulation was used (p = 0.216).

Notably, there was a substantial decrease in both median and mean adjudicated damages in the post-law school period (Table 1). The median award decreased by approximately 75.8% (from 12,682,749 KRW to 3,069,767 KRW), while the mean award decreased by about 39.0% (from 57,314,193 KRW to 34,980,344 KRW). Despite these apparent decreases, the Mann-Whitney U test indicated no statistically significant difference in damage awards between the two periods (p = 0.155), suggesting caution in interpreting these changes.

5. Liability Attribution Rate

Liability attribution rates were compared between two periods:

- 2006-2012 (n = 23):
  - Mean: 43.6% (SD = 38.33%)
  - Median: 50.0%
  - Range: 0% to 100%

- 2017-2021 (n = 75):
  - Mean: 37.0% (SD = 38.49%)
  - Median: 30.0%
  - Range: 0% to 100%

The analysis revealed a decrease in both mean and median liability attribution rates in the post-law school period (Table 3). The mean rate decreased by 6.6 percentage points (from 43.6% to 37.0%), while the median rate showed a more substantial decrease of 20 percentage points (from 50.0% to 30.0%). This suggests a trend towards lower liability attribution to physicians in more recent cases. However, a Mann-Whitney U test indicated no statistically significant difference in liability attribution rates between the pre-law school and post-law school periods (p = 0.539). The consistent range across both periods indicates that courts maintained a wide discretion (0% to 100%) in liability determinations throughout the study period.

6. Moral Damages Awards

- **2006–2012 (n = 23)**
  - Median: 15,000,000 KRW (11,217 USD)
  - Mean: 18,173,913 KRW (13,590 USD)
  - SD = 16,937,458 KRW (12,665 USD)
  - Range: 0 KRW to 50,000,000 KRW (0 USD to 37,389 USD)
  - Interquartile Range (IQR): 20,000,000 KRW (14,956 USD)

**Table 3. Distribution of Plaintiff Success Rates and Liability Attribution Before and After the Introduction of Law Schools.**

| Metric | Pre-Law School Period (2006–2012) | Post-Law School Period (2017–2021) | p-value |
|---|---|---|---|
| Plaintiff Success Rate (%) | 86.95 | 81.33 | 0.755 |
| Mean Liability Attribution (%) | 43.6 | 37 | – |
| Median Liability Attribution (%) | 50 | 30 | 0.539 |

•Plaintiff success rates were analyzed using Fisher's exact test and Liability attribution was analyzed using the Mann-Whitney U test.

- 2017-2021 (n = 75)
  - Median: 5,000,000 KRW (3,739 USD)
  - Mean: 13,060,000 KRW (9,766 USD)
  - SD = 23,150,209 KRW (17,311 USD)
  - Range: 0 KRW to 125,000,000 KRW (0 USD to 93,472 USD)
  - Interquartile Range (IQR): 12,000,000 KRW (8,973 USD)

Both periods exhibited right-skewed distributions, with the 2017–2021 period showing greater variability (higher SD) and a wider range of awards. The Mann-Whitney U test revealed a statistically significant difference in moral damages awards between the two periods (U = 601.000, p = 0.027).

The median award decreased from 11,217 USD in 2006–2012–3,739 USD in 2017–2021, suggesting a general trend towards lower moral damages awards in more recent cases. However, the wider range and higher standard deviation in the 2017–2021 period indicate increased variability in awards.

These findings suggest a shift in the judicial award patterns following the introduction of the law school system, with typically lower amounts but greater variability in recent years. This change may reflect evolving legal standards, judicial practices, or the nature of cases being litigated during this period.

7. Capital Region vs. Provincial Court Case Distribution: Pre- and Post-Law School System

This study analyzed the distribution of first-instance civil litigation cases before and after the implementation of the law school system in South Korea (Table 4). The analysis compared the pre-law school period with the post-law school period, examining changes in case proportions between courts located in the capital region (Seoul, Incheon, and Gyeonggi Province) and provincial areas.

The analysis revealed an increase in the proportion of first-instance civil litigation cases in capital region courts following the implementation of the law school system (Fig 1). Specifically, prior to the introduction of law schools, 13 cases were processed in capital region courts and 10 cases in provincial courts. After the introduction, the number of cases increased to 52 in capital region courts and 23 in provincial courts.

However, this change was not statistically significant. A chi-square test yielded a p-value of 0.255. This result indicates no statistically significant association between the implementation of the law school system and the distribution of cases between capital region and provincial courts.

**Table 4. Geographic Distribution and Case Duration of Malpractice Cases Before and After the Introduction of Law Schools.**

| Metric | Pre-Law School Period (2006–2012) | Post-Law School Period (2017–2021) | p-value |
|---|---|---|---|
| Capital Region Cases (%) | 56.52 | 69.33 | 0.255 |
| Provincial Cases (%) | 43.48 | 30.67 | 0.255 |
| Mean Case Duration (Days) | 1,016.74 | 1,594.68 | – |
| Median Case Duration (Days) | 969 | 1,570 | < 0.001 |

•The geographic distribution of cases was analyzed using the chi-square test, which showed no statistically significant difference between the two periods (p = 0.255).

•Case duration was analyzed using the Mann-Whitney U test, and the p-value indicates a statistically significant increase in the median duration in the post-law school period compared to the pre-law school period.

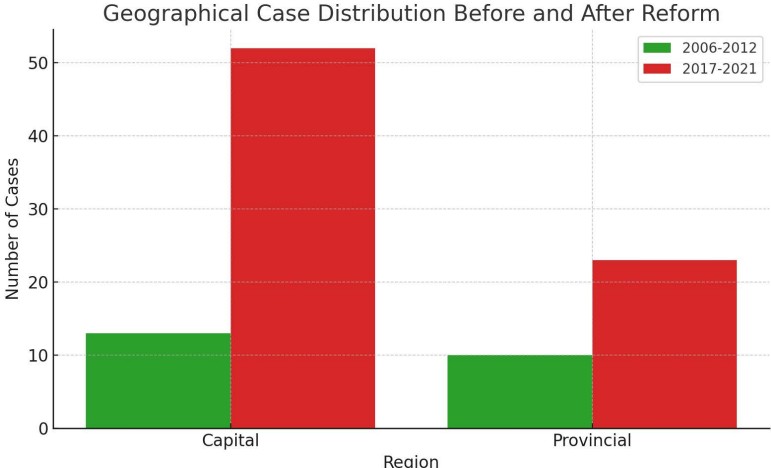

**Fig 1. Medical Malpractice Litigation in Plastic and Reconstructive Surgery: First Instance Civil Cases Before and After Law School Reform (2006-2012 vs. 2017-2021).** Chi-square test for regional distribution: p = 0.255.

8. Litigation Duration

The case duration was compared between the pre-law school (2006–2012) and post-law school (2017–2021) periods (Table 4):

- **2006–2012 period (n = 23):**

  - Median: 969 days

  - Mean: 1016.74 days (SD = 417.141 days)

  - Range: 487–2107 days

  - Interquartile Range (IQR): 571 days

- **2017–2021 period (n = 75):**

  - Median: 1570 days

  - Mean: 1594.68 days (SD = 758.453 days)

  - Range: 358–4235 days

  - Interquartile Range (IQR): 932 days

The Mann-Whitney U test revealed a statistically significant difference in case duration between the two periods (U = 1292.000, Z = 3.600, p < 0.001).

This analysis indicates a substantial increase in case duration from the pre-law school to the post-law school period (Fig 2). The median case duration increased by 601 days (from 969 to 1570 days), while the mean duration increased by approximately 578 days. The wider range and larger IQR in the post-law school period suggest greater variability in case durations during this time.

## Discussion

This study examines the impact of a significant reform in South Korea's legal education system (the introduction of law schools in 2009) on litigation related to PRSP, encompassing both surgical and non-surgical treatments. The reform was

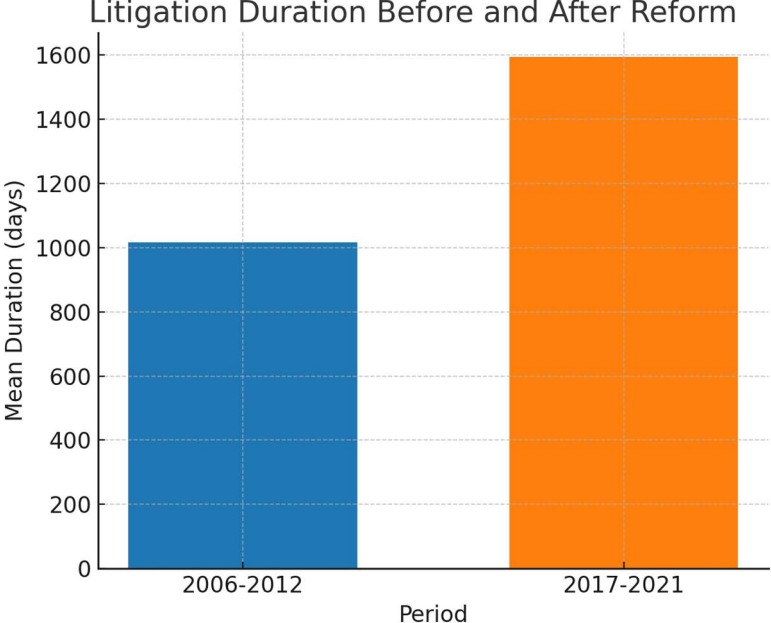

**Fig 2. Mean Duration of Medical Malpractice Litigation in PRSP Before and After Law School Reform (2006-2012: 1,011 days vs. 2017-2021: 1,592 days).** Due to non-normal distribution, Mann-Whitney U test was performed for median values: p < 0.001. PRSP: Plastic and Reconstructive Surgeries and Procedures.

designed to enhance the quality and specialization of legal professionals [18]. Our research presents one of the first empirical analyses of how this educational shift has influenced a specific area of medical law, focusing on malpractice cases related to plastic and reconstructive procedures.

## 1. Key findings and analysis

### 1.1. Increase in case frequency

A significant increase in the average annual number of PRSP lawsuits in the first instance was observed from the 2006–2012 period to the 2017–2021 period. This increase likely reflects a complex interplay of factors:

- Enhanced Legal Awareness: The introduction of law schools may have contributed to greater public understanding of legal rights in medical contexts, potentially lowering the threshold for litigation [19].

- Expansion of the Cosmetic Procedures Market: South Korea's growing cosmetic procedures industry naturally increases the potential for disputes [20].

- Increased Availability of Legal Services: The influx of law school graduates has likely made legal representation more accessible, possibly encouraging more patients to pursue litigation [21].

This increase in case frequency suggests that the legal education reform may have had broader societal impacts, potentially altering the dynamics between medical practitioners and patients. It raises important questions about the balance between patient rights, medical practice, and the role of legal professionals in mediating these relationships.

### 1.2. Increase in litigation duration

Our analysis revealed a statistically significant increase in case duration from a median of 969 days in the 2006–2012 period to 1,570 days in the 2017–2021 period (p < 0.001). This substantial increase of approximately 601 days can be explained through two distinct but interrelated factors:

**1.2.1. Fundamental changes in legal training system.** Prior to 2020, successful bar exam candidates underwent mandatory two-year practical training at the Judicial Research and Training Institute, providing intensive hands-on experience in litigation and legal practice [3]. The current law school system, which has completely replaced the previous system since 2020 [22], focuses primarily on academic legal education and bar exam preparation. Law school graduates can now practice immediately upon passing the bar exam, without the intensive practical training that was mandatory under the previous system [3].

This systemic change may affect the practical competency of new legal professionals, particularly in complex litigation such as medical malpractice cases.

Similar concerns emerged in Japan's post-2004 reform experience, where critics noted that the focus on bar exam preparation in law schools came at the expense of practical legal skills [23,24].

**1.2.2. System capacity constraints.** The increased number of legal professionals has not been matched by a corresponding increase in the number of judges and court infrastructure [25]. More accessible legal services have led to more diverse legal strategies and increased appeals, creating additional pressure on the existing judicial system. The combination of less experienced practitioners and limited judicial resources may contribute to longer resolution times

This paradoxical increase in litigation duration reflects the complex challenges of legal education reform. While the reform succeeded in increasing access to legal services, the shortened practical training period and unchanged judicial infrastructure may have contributed to longer case durations. Japan's similar experience following their 2004 reform, where increased litigation complexity coincided with concerns about practical training quality, provides an important parallel [26].

## 1.3. Patterns in adjudicated damages and liability attribution

Our analysis revealed parallel decreases in both damages and liability attribution during the post-law school period. The median damage award decreased by 75.8%, and the mean by 39%, while liability attribution rates showed reductions in both mean (43.6% to 37.0%) and median (50.0% to 30.0%) values. However, Mann-Whitney U tests indicated no statistically significant differences between periods for either damages (p = 0.155) or liability attribution (p = 0.539).

Two interconnected factors likely underpin these trends. First, lawsuits related to minimally invasive procedures, particularly filler and botulinum toxin treatments, have increased significantly [18]. While these cases include severe complications such as skin necrosis and even blindness, many involve outcomes that may not result in long-term harm but are nonetheless distressing to patients, such as localized tissue damage or temporary impairments [18].

Second, the growing number of law school graduates has reduced attorney fees [27], making legal representation more accessible and increasing litigation of cases that might have been informally resolved in the past.

Although liability attribution rates maintained a wide and consistent range (0% to 100%) across both periods, the observed decreases in both damages and attribution rates suggest evolving litigation dynamics. These trends may reflect shifts in case characteristics, the severity of claims brought to court, or changing legal strategies. Further analysis is warranted to explore how these factors interact and shape judicial outcomes in medical litigation.

## 1.4. Emergence of new procedure types in litigation

This study revealed a shift in the types of procedures involved in malpractice litigation. Cases involving laser and ultrasound procedures (10.6%) and facial bone fracture reconstruction (4%) emerged in the 2017–2021 period, while absent in 2006–2012.

This change may be attributed to:

1. Technological Advancements: Increased use of laser and ultrasound procedures in cosmetic treatments.

2. Complexity of Reconstructive Procedures: Facial bone fracture reconstruction cases involve potentially severe complications.

 

3. Evolving Legal Landscape: The new law school system may have broadened understanding of medical malpractice law.

4. Changing Patient Expectations: Increased expectations for procedure outcomes.

It's unclear whether this represents an overall increase in litigation for these procedures or a shift in case types reaching courts. This trend underscores the need for legal and medical professionals to stay informed about emerging medical technologies and techniques.

### 1.5. Societal context and cultural influences

The observed trends in PRSP litigation must be understood within South Korea's unique societal context. South Korea has the world's highest per capita rate of plastic surgery procedures, with recent statistics showing that approximately one-third of women aged 19–29 have undergone cosmetic procedures [28]. This high prevalence reflects deeper cultural attitudes toward physical appearance and its perceived role in social and professional success.

Several cultural factors distinctively influence litigation patterns in this field. First, the normalization of plastic surgery in Korean society has led to heightened expectations for surgical outcomes. Unlike many other countries where plastic surgery is viewed primarily as a medical or corrective procedure, Korean society has widely embraced these procedures as routine investments in self-improvement [29], leading to more assertive responses when outcomes fall short of expectations. This cultural acceptance may partially explain the increase in litigation frequency observed in our study.

However, this trend is moderated by traditional Korean cultural values that historically discouraged litigation. The Confucian emphasis on harmony and conflict avoidance has traditionally made Koreans reluctant to pursue legal action [30]. The observed increase in litigation rates thus suggests a significant shift in cultural attitudes, potentially accelerated by the increased accessibility to legal services following the law school reforms [27]. This cultural transition is particularly evident in younger generations, who show greater willingness to assert their rights through legal channels. PRSP litigation in South Korea reflects a balance between traditional values and modern expectations. While patients are increasingly pursuing legal action, the stability in liability attribution rates likely stems from consistent legal standards, despite evolving societal attitudes.

Social media and South Korea's highly digitalized society have introduced new dimensions to plastic surgery litigation. The widespread sharing of surgical outcomes online has increased transparency, enabling patients to make more informed choices [31]. However, it has also elevated expectations for perfect results, leading to greater dissatisfaction when outcomes fall short. This dynamic is amplified by South Korea's 'digital beauty culture,' where the normalization and public scrutiny of cosmetic procedures heighten social pressures and contribute to the complexity and frequency of litigation. Social media platforms, as outlets for public dissatisfaction, further influence legal disputes and pose reputational risks for medical providers [32].

## 2. Legal and international implications

The relationship between legal education systems and their impact on medical litigation efficiency varies significantly across jurisdictions. South Korea's Legal Education Eligibility Test (LEET), introduced alongside the law school system in 2009, differs markedly from entrance requirements in other countries. While the LEET emphasizes analytical and logical reasoning similar to the United States' Law School Admission Test (LSAT), it includes distinct components reflecting Korea's unique legal needs, particularly in language proficiency and social science knowledge [25]. This distinctive approach to legal education assessment may influence how future legal professionals handle complex medical litigation.

Expanding judicial capacity is essential for addressing the growing litigation burden in South Korea, particularly in the complex field of PRSP litigation [18]. While increasing the number of judges and establishing specialized court divisions are important steps, integrating specialized judicial systems and advanced technologies offers transformative potential [18].

The United States has long debated the establishment of medical courts as a solution to its malpractice litigation crisis [33]. Proposed models involve replacing traditional juries with expert panels trained in medical law and practice, aiming to provide fairer and more consistent decisions in approximately half the time of jury trials [34]. Such courts are expected to reduce costs, improve outcomes, and ensure greater predictability in medical malpractice disputes [34].

Japan's experience following its 2004 legal education reform offers particularly relevant insights for South Korea. Despite initial similarities to the Korean model, Japan maintained its Legal Training and Research Institute, requiring substantial practical training [35]. Furthermore, Japan's specialized medical litigation departments within major district courts demonstrate how institutional specialization can enhance efficiency in handling complex medical cases [36].

Germany offers a well-established model for resolving medical disputes outside the court system through *Gutachterkommissionen* (expert review panels) and *Schlichtungsstellen* (mediation offices) [37]. These independent bodies, composed of medical and legal experts, provide impartial evaluations of alleged medical malpractice cases. By assessing the presence of medical errors and their impact on patients, these panels aim to resolve disputes efficiently and fairly [37].

One of the key strengths of this system is its ability to significantly reduce the burden on the judicial system. According to the German Medical Association, over 90% of medical malpractice cases handled by these panels are resolved without proceeding to litigation [11]. This approach not only shortens resolution times - most cases are resolved within a year - but also minimizes costs for all parties involved, as the system is largely funded by professional associations [11].

The voluntary nature of these procedures ensures that both parties must agree to participate, fostering a cooperative environment. Furthermore, the use of standardized processes and centralized reporting systems, such as the Medical Error Reporting System (MERS), enhances transparency and builds trust among patients and healthcare providers [38]. These features make Germany's system an exemplary model for balancing efficiency, fairness, and accessibility in medical dispute resolution.

These international comparisons suggest that maintaining robust practical training components alongside academic education might better prepare legal professionals for complex medical litigation. Moving forward, South Korea could benefit from considering elements of these systems, particularly Japan's specialized medical litigation departments and Germany's alternative dispute resolution mechanisms, while preserving the unique characteristics of its own legal education system.

China's Smart Court initiative exemplifies how technology can enhance judicial efficiency. Since its introduction in 2017, the system has utilized artificial intelligence (AI) to automate case management, analyze legal documents, and generate judicial recommendations based on prior rulings [39,40]. Blockchain technology ensures evidence authentication, while online litigation platforms streamline procedural tasks [41]. Between 2019 and 2021, innovations in China's smart court system significantly reduced judges' workload by approximately 40% in some courts, improved trial efficiency by around 30%, and enhanced overall judicial system productivity [40]. These advancements contributed to measurable reductions in administrative burdens and operational delays, leading to improved access to justice and streamlined case management.

While China's Smart Court exemplifies technological integration in judicial processes, the United Kingdom and Singapore offer valuable models for leveraging technology in legal education specifically focused on medical litigation training.

The United Kingdom has pioneered specialized technology-enhanced learning environments for medical litigation education. British institutions employ Learning Management Systems (LMS) and AI-powered legal research tools to develop specialized competencies in medical law. According to recent research [42], these digital platforms enhance accessibility and learning flexibility while developing crucial practical skills. The UK approach emphasizes Virtual Reality simulations of medical malpractice proceedings, where law students engage with complex medical evidence and expert testimony in immersive courtroom environments. This technology enables future lawyers to develop specialized advocacy skills for medical malpractice cases before entering actual courtrooms.

Singapore's legal education system has integrated digital technologies through a comprehensive approach focused on practical application. Singapore law schools utilize online collaboration tools that foster deeper engagement with medical litigation concepts and develop argumentation skills specifically for medical malpractice contexts [43]. Their phased technology adoption strategy ensures proper training and adjustment, minimizing disruption to the learning process while maintaining educational quality. Singapore's system particularly emphasizes the use of AI-powered legal research tools that improve students' research efficiency and depth of analysis in medical malpractice cases by automating certain aspects of legal research [44].

The technology-enhanced educational approaches from the UK and Singapore demonstrate how digital integration in legal education can specifically address the procedural inefficiencies currently observed in South Korea's PRSP litigation. By implementing similar specialized medical litigation education technologies with clear pedagogical alignment and comprehensive support systems, South Korea could bridge the gap between theoretical knowledge and practical application in medical malpractice litigation while maintaining the core tenets of legal education.

Collectively, the international examples from China, the United Kingdom, and Singapore underscore the importance of a multi-faceted approach to judicial reform. A phased strategy is recommended for South Korea. In the short term, increasing judicial resources, such as appointing additional judges and creating specialized divisions for medical litigation, should be prioritized. In the mid-term, adopting technology-driven solutions can further streamline judicial processes - both through China's Smart Court-style case management systems and by implementing UK and Singapore's legal education technology models to better prepare new lawyers for efficient handling of medical malpractice cases. Finally, in the long term, integrating specialized court models, such as medical courts in the United States or pre-trial expert assessments in Germany, alongside comprehensive technology-enhanced legal education programs, can provide sustainable solutions for managing the increasing complexity of PRSP litigation.

## 3. Limitations and future research directions

This study has several limitations that should be addressed in future research:

1. Sample Size: The relatively small sample size, particularly in the pre-law school period (n = 23), limits the statistical power of our analyses. Future studies with larger datasets could provide more robust and generalizable findings. However, the availability of relevant court cases constrains sample size. This limitation should be acknowledged, and future research should strive to include additional cases as they become available.

2. Geographical Limitation: This study is centered on South Korea, and its findings may not be fully applicable to other legal systems. Comparative analysis with nations like the United States and Japan, which have implemented similar legal education reforms, could offer broader insights into both unique and shared impacts.

3. Influential Factors and Causality: This study did not fully account for factors such as advancements in medical technology, shifting societal attitudes towards plastic and reconstructive procedures, and broader legal reforms. These uncontrolled variables may have influenced the observed trends, complicating efforts to establish a direct causal relationship between legal education reform and changes in litigation patterns. Future research should aim to isolate these effects by incorporating data on additional variables and employing methodologies better suited to establishing causality.

4. Data Exclusions: The study excluded cases from 2013 to 2017 to ensure that the post-law school period fully reflected the adaptation of law school graduates into the legal profession. This exclusion was necessary not only to capture the substantive effects of the reform after new judges had established their autonomous judicial identity, but also because the number of law school graduates in the legal profession during this period was extremely limited, making their impact on litigation patterns and judicial decisions negligible. While this methodological choice may introduce potential bias in my analysis, the practical insignificance of law school graduates' presence during the transitional

period supports this exclusion decision. Specifically, this transitional period, although potentially containing valuable insights into institutional adjustment processes, would likely yield insufficient data regarding the reform's impact on actual litigation and judicial decision-making due to the minimal presence of law school graduates in the profession. Future research should examine this transitional period to provide a more comprehensive understanding of the reform's effects, particularly focusing on how the reform's impact evolved from its initial implementation to its mature stage, when the number of law school graduates in the legal profession reached a meaningful threshold for analysis.

5. Limited Scope of Analysis: This study focuses exclusively on plastic and reconstructive surgery-related malpractice cases. While this provides a unique lens into the legal and medical systems, the findings may not be generalizable to other medical fields or types of legal disputes. Broader analyses covering diverse legal contexts could enrich our understanding of the reform's broader implications.

6. Small Claims Exclusion: Our study necessarily excluded small claims court decisions due to significant practical constraints in the South Korean judicial system [45]. Small claims cases, which are typically adjudicated by a single judge under considerable time pressure and heavy caseload conditions, rarely produce detailed written judgments [45]. Based on insights from practicing judges, the overwhelming workload associated with small claims adjudication results in minimal documentation of legal reasoning. While this exclusion was methodologically unavoidable due to the lack of accessible written judgments, we acknowledge that these cases might represent a distinct subset of medical disputes, particularly those involving minor procedures or partial complaints. However, given the typically high-value nature of PRSP malpractice claims and the complexity of procedures involved, we believe our dataset captures the most substantively significant cases in this field. The exclusion of small claims cases, while limiting the comprehensiveness of the analysis, likely does not substantially alter our main findings due to their relatively limited complexity and scope. It is important to note that the limited documentation in small claims courts is a structural feature of the South Korean judicial system, reflecting the inherent time constraints and efficiency demands of handling high-volume, lower-stakes cases. This practice has remained consistent historically and, given the institutional priorities and resource allocation in the judicial system, is likely to persist. Therefore, rather than suggesting potential future inclusion of these cases, we acknowledge this as a permanent characteristic of our legal system that shapes the boundaries of empirical legal research in this field.

Despite these limitations, this study provides critical empirical insights into the unintended consequences of South Korea's legal education reform. It highlights the importance of aligning systemic reforms with judicial capacity and offers valuable lessons for both policymakers and researchers. Future studies should aim to address these limitations to build a more comprehensive understanding of the relationship between legal education reforms and systemic outcomes.

## Conclusions

This study provides empirical evidence of the impact of South Korea's legal education reform on plastic and reconstructive surgery-related litigation. Key findings include a significant increase in both the frequency and duration of malpractice cases following the introduction of law schools, while adjudicated damages and liability attribution rates showed downward trends, albeit not statistically significant. These results reflect both the procedural consequences of the reform, particularly the growing litigation caseload and extended case durations, and subtle shifts in case characteristics.

The increase in litigation frequency aligns with the original intent of the law school system, which aimed to enhance accessibility to legal representation and empower individuals to pursue their rights through formal legal channels. However, the observed delays in case resolutions highlight systemic challenges in managing judicial efficiency amid rising litigation demands. While the observed decreases in adjudicated damages and liability attribution were not statistically significant, suggesting courts' overall consistency in substantive decision-making, the extended procedural timelines underscore the need for system-level adaptations to handle increased complexity.

To address these challenges, policymakers should focus on enhancing judicial capacity. Short-term measures include appointing additional judges and establishing specialized court divisions for medical disputes. In the mid-term, adopting AI-based case management systems could streamline procedural workflows and improve efficiency. Long-term goals should prioritize the development of alternative dispute resolution mechanisms, such as structured mediation programs, to alleviate the burden on courts and provide timely resolutions.

This study not only sheds light on South Korea's unique challenges but also offers valuable lessons for other countries undergoing similar legal education reforms. Future research should explore the broader effects of these reforms on other areas of medical litigation and investigate comparative approaches in countries such as Japan, Germany, and China. Additionally, understanding the perspectives of legal and medical professionals will be critical for designing reforms that effectively balance judicial efficiency and fairness.

## Supporting information

**S1 File. Court decision analysis data for plastic and reconstructive surgery malpractice cases. This dataset contains the detailed analysis of court decisions used in this study, including case details, damages awarded, and duration of litigation.**
(XLSX)

## Author contributions

**Conceptualization:** Daihun Kang.

**Data curation:** Daihun Kang.

**Formal analysis:** Daihun Kang.

**Funding acquisition:** Daihun Kang.

**Investigation:** Daihun Kang.

**Methodology:** Daihun Kang.

**Project administration:** Daihun Kang.

**Resources:** Daihun Kang, Seung Eun Hong.

**Software:** Daihun Kang.

**Supervision:** Daihun Kang, Seung Eun Hong.

**Validation:** Daihun Kang, Seung Eun Hong.

**Visualization:** Daihun Kang.

**Writing – original draft:** Daihun Kang.

**Writing – review & editing:** Daihun Kang, Seung Eun Hong.

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
