## [Decision Letter · Decision Letter 0]

3 Jan 2025

PONE-D-24-54910South Korea’s Legal Education Reform: More Lawyers, More Lawsuits, but Longer DelaysPLOS ONE

Dear Dr. Kang,

Thank you for submitting your manuscript to PLOS ONE. After careful consideration, we feel that it has merit but does not fully meet PLOS ONE’s publication criteria as it currently stands. Therefore, we invite you to submit a revised version of the manuscript that addresses the points raised during the review process.

The article is scientifically valid but needs implementation, as indicated by reviewers.

We look forward to receiving your revised manuscript.

Kind regards,

Andrea Cioffi

Academic Editor

PLOS ONE

Additional Editor Comments (if provided):

Reviewers' comments:

Reviewer's Responses to Questions

**Comments to the Author**

1. Is the manuscript technically sound, and do the data support the conclusions?

Reviewer #1: Yes

Reviewer #2: Yes

Reviewer #3: Partly

2. Has the statistical analysis been performed appropriately and rigorously? 

Reviewer #1: N/A

Reviewer #2: Yes

Reviewer #3: No

3. Have the authors made all data underlying the findings in their manuscript fully available?

Reviewer #1: Yes

Reviewer #2: Yes

Reviewer #3: Yes

4. Is the manuscript presented in an intelligible fashion and written in standard English?

Reviewer #1: Yes

Reviewer #2: Yes

Reviewer #3: No

5. Review Comments to the Author

Reviewer #1: General comments:

Based on focus of the paper, the paper aligns with PLOS ONE's scope, offering an empirical analysis of legal education reform and its unintended consequences on malpractice litigation in South Korea. The research provides valuable insights, is well-structured, and uses appropriate statistical methods. However, the paper needs improvements in contextual analysis, causality interpretation, and broader applicability of findings before it becomes ready for publication

Comments to be address by the author before acceptence:

1. The abstract lacks details on causative factors and practical implications. Kindly revise the abstract to include more actionable insights for policymakers and practitioners.

2. The statistical methods (Mann-Whitney U test, Fisher’s Exact Test) are appropriate. However, the manuscript should clearly explain the selection of statistical tests for each metric (e.g., why Monte Carlo simulations were needed for specific Fisher's tests). So please provide additional justification for the use of non-parametric tests for damages and liability attribution data?

3. While the data is detailed, adding more visual elements (e.g., trend graphs, heat maps) would make trends easier to interpret. Could you include graphical representations to highlight trends in litigation duration and geographical case distribution?

4. The study excludes cases from 2013–2017. While justified, this exclusion could create a potential gap in understanding the immediate impacts of the reform. Please explain how do you ensure that excluding data from this transitional period does not introduce bias? Would including some of these cases provide additional insights?

5. The exclusion of small claims court decisions might omit a significant subset of medical disputes. Could the exclusion of small claims cases significantly affect the representativeness of your findings?

6. The increase in litigation duration is well-documented but requires more discussion on potential causes (e.g., systemic inefficiencies, strategic legal approaches). So expand on why litigation durations have increased so significantly despite the influx of legal professionals?

7. The findings focus on South Korea but lack direct comparisons to similar legal reforms globally. Need comparison of South Korea’s litigation trends to those in other countries with similar legal reforms (e.g., Japan and US) to enhance the paper’s international relevance.

8. The policy suggestions (e.g., increasing judicial capacity, introducing alternative dispute mechanisms) are relevant but lack specifics or examples. Please provide case studies or examples of similar policy interventions in other jurisdictions to support your recommendations.

9. The role of South Korea’s cultural attitudes toward plastic surgery and litigation is briefly mentioned but not deeply analyzed. Please tell us how might societal perceptions of cosmetic surgery influence litigation trends?

10. The manuscript is well-written but includes technical terms (e.g., “Monte Carlo simulation”) that may be unfamiliar to a broader audience. Please simplify or explain technical terms to ensure accessibility for a wider readership.

Publishable with Major Revisions: The manuscript has potential, but it needs improvements in contextual analysis, comparative perspectives and clarity. If the author addresses the above comments, the paper could be a valuable contribution to PLOS ONE.

Reviewer #2: Comments/ Observations on Manuscript titled:

South Korea’s Legal Education Reform: More Lawyers, More Lawsuits, but Longer

Delays (Manuscript Number: PONE-D-24-54910)

The following are the comments/observations:

1. The abstract should capture methodology of the study.

2. The introduction is not so explicitly elaborated to elicit various variables of the study.

3. It is important to elucidate on the necessity of legal education reform in South-Korea, and explain the conceptualization of “more lawyers”, “more lawsuits” and “longer delays”.

4. The introduction should explain the rationale for the study, the gap to be filled and the novelty of the study.

5. More literature review should be added to introduction since there is no subheading for literature review part. The following should be considered regarding the review of more studies:

i. Explain a transition from a largely apprenticeship-based system to a more structured graduate-level education model in South-Korea.

ii. Concerns about quality which arose about inconsistencies in the quality of legal education and preparedness of graduates from the realities of legal practices should be elaborated in details.

iii. Standardized curriculum which reform was introduced covering core legal subjects should be explicated.

iv. Explain competitive entrance examination (LEET) which was introduced to select candidate for law school based on aptitude and academic achievement.

v. Explain current challenges and future direction.

6. Elaborate on how leveraging technology into legal education can enhance efficiency and learning outcomes in South-Korea as in the cases of UK and Singapore.

7. The researcher (s) should explore legal education reform from some selected countries such as: US, UK, Canada through legal education reform in South-Korea can learn from different lessons.

8. Regarding ethical considerations, the researcher (s) posited that: “This study was conducted in compliance with ethical standards for research”. Explain what you mean by ethical standards for research and justify with literature.

9. Under “Data Analysis”, the researcher should write the full meaning of SPSS as shown the paper.

10. Tables1 to 4 should be incorporated into the appropriate places in the contents of the paper.

11. The organization and arrangement of the paper should align with the standard of the journal.

12. The results should be elaborated using various studies to justify the overall findings of the study.

13. Recheck typographic, spelling and grammatical errors.

14. By incorporating all the foregoing, it will improve the quality of the paper in order to make legal education reforms to be tailored to South-Korea’s specific context, legal culture and improvement of societal needs.

Reviewer #3: The study provides valuable insights into the unintended consequences of South Korea’s legal education reform. It has significant value and potential but requires substantial improvements in methodology, analysis, and interpretation.

The methodology includes a retrospective analysis with clear pre- and post-reform time frames. The selection of publicly available court decisions aligns with the study's objectives. The criteria for case inclusion and exclusion are clearly defined, ensuring consistency and clarity in the dataset. However, broader legal and healthcare reforms during the study period may have influenced the findings, but these factors were not thoroughly explored. Furthermore, the pre-reform sample size is relatively small (n=23), limiting the statistical power of comparisons and potentially affecting the study's generalizability and robustness.

While the study identifies correlations, it does not establish a clear causal link between the reform and litigation trends. Without stronger causal evidence, the findings may not fully support the conclusions. The weaknesses in the methodology directly impact the reliability of the findings, necessitating significant revisions to address these gaps before the study’s claims can be confidently supported.

The discussion is moderately well-developed but requires substantial refinement to fully capitalize on the findings and contextualize them within broader academic and policy debates. While it effectively summarizes the results and offers high-level insights, the lack of depth and specificity undermines its impact. For instance, the discussion references some studies but does not engage deeply with the broader academic discourse on legal education reform or medical malpractice litigation. A more thorough comparison with international trends or prior studies could strengthen the discussion. Furthermore, integrating the findings with existing studies on legal education reform and medical litigation trends, particularly in South Korea and other countries with similar reforms, is recommended.

Additionally, deeper insights into the mechanisms driving the increase in litigation frequency and duration are needed. While these trends are attributed to broad factors such as greater public awareness and increased availability of legal services, the discussion lacks detailed explanations of these changes. Some conclusions overstate the findings, such as attributing judicial inefficiencies entirely to the reform without sufficient evidence. The recommendations are also broad and lack actionable specificity, such as how to implement alternative dispute resolution mechanisms or expand judicial capacity.

In short, the paper addresses an underexplored topic—the unintended consequences of legal education reform on medical malpractice litigation—which is valuable for both legal and medical disciplines, particularly given South Korea's unique context as a global leader in plastic surgery. The research provides a foundation that can be further developed.

However, the weaknesses in methodology, data analysis, discussion, and scope significantly undermine its reliability and generalizability. Additionally, the paper would benefit from language proofreading and editing. These issues require substantial revisions to strengthen the study’s contribution to academic and policy discourse.

6. PLOS authors have the option to publish the peer review history of their article (what does this mean? ). If published, this will include your full peer review and any attached files.

**Do you want your identity to be public for this peer review?** For information about this choice, including consent withdrawal, please see our Privacy Policy .

Reviewer #1: No

Reviewer #2: **Yes: ** Dr. Yusuff Jelili Amuda

Reviewer #3: No

---

## [Author Response · Author response to Decision Letter 1]

19 Feb 2025

We sincerely appreciate the reviewers' thoughtful and constructive feedback, which has been instrumental in improving our manuscript. As medical researchers accustomed to writing within the IMRaD framework and typical word limits of 2,500-3,000 words, some suggestions - particularly those requesting extensive background information on legal education reforms - posed unique challenges. Nevertheless, we have made substantial efforts to address the reviewers' concerns while maintaining the manuscript's scientific rigor and accessibility.

In response to specific comments, we have: enhanced the methodology section with clearer justification for statistical methods; expanded the discussion to include international comparisons with Japan, China, and Germany; addressed the limitations of our sample size and retrospective design; elaborated on the practical constraints regarding small claims court data in South Korea; and added more specific, actionable recommendations.

We believe these revisions have significantly strengthened the manuscript's depth, clarity, and relevance to both medical and legal audiences.

---

## [Decision Letter · Decision Letter 1]

10 Mar 2025

PONE-D-24-54910R1Legal Education Reform and Medical Litigation: Improved Access but Delayed Justice in Plastic Surgery Malpractice Cases in South KoreaPLOS ONE

Dear Dr. Hong,

Thank you for submitting your manuscript to PLOS ONE. After careful consideration, we feel that it has merit but does not fully meet PLOS ONE’s publication criteria as it currently stands. Therefore, we invite you to submit a revised version of the manuscript that addresses the points raised during the review process.

We look forward to receiving your revised manuscript.

Kind regards,

Andrea Cioffi

Academic Editor

PLOS ONE

Journal Requirements:

Reviewers' comments:

Reviewer's Responses to Questions

**Comments to the Author**

1. If the authors have adequately addressed your comments raised in a previous round of review and you feel that this manuscript is now acceptable for publication, you may indicate that here to bypass the “Comments to the Author” section, enter your conflict of interest statement in the “Confidential to Editor” section, and submit your "Accept" recommendation.

Reviewer #1: All comments have been addressed

Reviewer #2: All comments have been addressed

Reviewer #3: All comments have been addressed

2. Is the manuscript technically sound, and do the data support the conclusions?

Reviewer #1: Yes

Reviewer #2: Yes

Reviewer #3: Yes

3. Has the statistical analysis been performed appropriately and rigorously? 

Reviewer #1: Yes

Reviewer #2: Yes

Reviewer #3: Yes

4. Have the authors made all data underlying the findings in their manuscript fully available?

Reviewer #1: Yes

Reviewer #2: Yes

Reviewer #3: Yes

5. Is the manuscript presented in an intelligible fashion and written in standard English?

Reviewer #1: Yes

Reviewer #2: Yes

Reviewer #3: Yes

6. Review Comments to the Author

Reviewer #1: (No Response)

Reviewer #2: The following are the comments/observations:

Comment 1: The abstract should capture methodology of the study.

Response: It is done and okay.

Comment 2: The introduction is not so explicitly elaborated to elicit various variables of the study.

Response: It has been addressed as suggested.

Comment 3: It is important to elucidate on the necessity of legal education reform in South-Korea, and explain the conceptualization of “more lawyers”, “more lawsuits” and “longer delays”.

Response: The title has been modified and the content is aligned with the title of the manuscript.

Comment 4: The introduction should explain the rationale for the study, the gap to be filled and the novelty of the study.

Response: It has been done as suggested.

Comment 5: More literature review should be added to introduction since there is no subheading for literature review part. The following should be considered regarding the review of more studies:

i. Explain a transition from a largely apprenticeship-based system to a more structured graduate-level education model in South-Korea.

ii. Concerns about quality which arose about inconsistencies in the quality of legal education and preparedness of graduates from the realities of legal practices should be elaborated in details.

iii. Standardized curriculum which reform was introduced covering core legal subjects should be explicated.

iv. Explain competitive entrance examination (LEET) which was introduced to select candidate for law school based on aptitude and academic achievement.

v. Explain current challenges and future direction.

Response: It All the abovementioned have been explained in the content as pointed out.

Comment 6: Elaborate on how leveraging technology into legal education can enhance efficiency and learning outcomes in South-Korea as in the cases of UK and Singapore.

Response: This part is not addressed as suggested and no justifiable reason is given for not addressing it.

Comment 7: The researcher (s) should explore legal education reform from some selected countries such as: US, UK, Canada through legal education reform in South-Korea can learn from different lessons.

Response: This has addressed in the content in p. 38.

Comment 8: Regarding ethical considerations, the researcher (s) posited that: “This study was conducted in compliance with ethical standards for research”. Explain what you mean by ethical standards for research and justify with literature.

Response: It has been addressed as suggested.

Comment 9: Under “Data Analysis”, the researcher should write the full meaning of SPSS as shown the paper.

Response: It has been addressed in p. 16 as suggested.

Comment 10: Tables1 to 4 should be incorporated into the appropriate places in the contents of the paper.

Response: It has been done.

Comment 11: The organization and arrangement of the paper should align with the standard of the journal.

Response: This should be check by the editor or editor in chief.

Comment 12: The results should be elaborated using various studies to justify the overall findings of the study.

Response: It has been addressed as pointed out.

Comment 13: Recheck typographic, spelling and grammatical errors.

Response: It has been improved as suggested.

Reviewer #3: (No Response)

7. PLOS authors have the option to publish the peer review history of their article (what does this mean? ). If published, this will include your full peer review and any attached files.

**Do you want your identity to be public for this peer review?** For information about this choice, including consent withdrawal, please see our Privacy Policy .

Reviewer #1: No

Reviewer #2: No

Reviewer #3: **Yes: ** Muayad K Hattab

---

## [Author Response · Author response to Decision Letter 2]

11 Mar 2025

Comment 6: Elaborate on how leveraging technology into legal education can enhance efficiency and learning outcomes in South-Korea as in the cases of UK and Singapore.

Response: We thank the reviewer for this valuable suggestion. We have added a comprehensive discussion of how the UK and Singapore leverage technology in legal education to enhance efficiency and learning outcomes, particularly in the context of medical litigation training. This addition can be found in the "Legal and International Implications" section. We have described the UK's use of Learning Management Systems and Virtual Reality simulations for medical malpractice proceedings, as well as Singapore's comprehensive approach to digital technology integration through online collaboration tools and AI-powered legal research tools. We have also discussed how these approaches could be applied to address procedural inefficiencies in South Korea's PRSP litigation. These additions are clearly marked in red text in the revised manuscript with track changes, making them easy to identify.

---

## [Decision Letter · Decision Letter 2]

21 Mar 2025

Legal Education Reform and Medical Litigation: Improved Access but Delayed Justice in Plastic Surgery Malpractice Cases in South Korea

PONE-D-24-54910R2

Dear Dr. Seungeun Hong,

We’re pleased to inform you that your manuscript has been judged scientifically suitable for publication and will be formally accepted for publication once it meets all outstanding technical requirements.

Kind regards,

Massimo Finocchiaro Castro, PhD

Academic Editor

PLOS ONE

Additional Editor Comments (optional):

Reviewers' comments:

Reviewer's Responses to Questions

**Comments to the Author**

1. If the authors have adequately addressed your comments raised in a previous round of review and you feel that this manuscript is now acceptable for publication, you may indicate that here to bypass the “Comments to the Author” section, enter your conflict of interest statement in the “Confidential to Editor” section, and submit your "Accept" recommendation.

Reviewer #1: All comments have been addressed

Reviewer #2: All comments have been addressed

2. Is the manuscript technically sound, and do the data support the conclusions?

Reviewer #1: Yes

Reviewer #2: Yes

3. Has the statistical analysis been performed appropriately and rigorously? 

Reviewer #1: Yes

Reviewer #2: Yes

4. Have the authors made all data underlying the findings in their manuscript fully available?

Reviewer #1: Yes

Reviewer #2: Yes

5. Is the manuscript presented in an intelligible fashion and written in standard English?

Reviewer #1: Yes

Reviewer #2: Yes

6. Review Comments to the Author

Reviewer #1: (No Response)

Reviewer #2: The following are the comments/observations:

Comment 1: Elaborate on how leveraging technology into legal education can enhance efficiency and learning outcomes in South-Korea as in the cases of UK and Singapore.

Response: This part has been addressed as suggested.

7. PLOS authors have the option to publish the peer review history of their article (what does this mean? ). If published, this will include your full peer review and any attached files.

**Do you want your identity to be public for this peer review?** For information about this choice, including consent withdrawal, please see our Privacy Policy .

Reviewer #1: **Yes: ** Asif Kamal

Reviewer #2: **Yes: ** Associate Prof. Dr. Yusuff Jelili Amuda

---

## [Editor Report · Acceptance letter]

PONE-D-24-54910R2

PLOS ONE

Dear Dr. Hong,

I'm pleased to inform you that your manuscript has been deemed suitable for publication in PLOS ONE. Congratulations! Your manuscript is now being handed over to our production team.

Kind regards,

on behalf of

Prof. Massimo Finocchiaro Castro

Academic Editor

PLOS ONE